# Postbiotics—A Step Beyond Pre- and Probiotics

**DOI:** 10.3390/nu12082189

**Published:** 2020-07-23

**Authors:** Jakub Żółkiewicz, Aleksandra Marzec, Marek Ruszczyński, Wojciech Feleszko

**Affiliations:** 1Department of Pediatric Pulmonology and Allergy, Medical University of Warsaw, Żwirki i Wigury 63A, 02-091 Warsaw, Poland; kuba.zolkiewicz@gmail.com (J.Ż.); aleksandraxmarzec@gmail.com (A.M.); 2Department of Paediatrics, Medical University of Warsaw, Żwirki i Wigury 63A, 02-091 Warsaw, Poland; marek.ruszczynski@gmail.com

**Keywords:** bacteria, inflammation, microbiome, postbiotic, prebiotic, probiotic

## Abstract

As an imbalance in the intestinal microbiota can lead to the development of several diseases (e.g., type 1 diabetes, cancer, among others), the use of prebiotics, probiotics, and postbiotics to alter the gut microbiome has attracted recent interest. Postbiotics include any substance released by or produced through the metabolic activity of the microorganism, which exerts a beneficial effect on the host, directly or indirectly. As postbiotics do not contain live microorganisms, the risks associated with their intake are minimized. Here, we provided a critical review of postbiotics described in the literature, including their mechanisms of action, clinical characteristics, and potential therapeutic applications. We detailed the pleiotropic effects of postbiotics, including their immunomodulatory, anti-inflammatory, antioxidant, and anti-cancer properties. Although the use of postbiotics is an attractive strategy for altering the microbiome, further study into its efficacy and safety is warranted.

## 1. Introduction

The assemblage of microorganisms that inhabit the human body, their genomes and metabolites, as well as the environment in which they live, is called the microbiota. Microorganisms that are part of the microbiome can be isolated from all areas in constant contact with the external environment (e.g., the skin, upper respiratory tract, or urogenital tract). However, they are most abundant in the gastrointestinal tract. Our interdependent relationship with the intestinal microbiota is established during the first three years of life [1]. The human body provides a stable, nutrient-rich environment for the inhabiting microorganisms, and in return, receives a number of benefits. These benefits include stimulation of the immune system, improved digestion and absorption of food, reduced growth of pathogenic flora, and maintenance of intestinal barrier integrity. These beneficial effects of the interaction between the microbiota and the gastrointestinal tract can be observed not only locally, but also in distant organs, due to systemic distribution of substances and cells produced in the intestine. This phenomenon is called the gut-organ axis, according to which we can distinguish the gut-brain, gut-skin, gut-lung axis, and so on.

Several factors can affect the composition of the microbiota starting from the perinatal period, including the composition of the maternal gut microbiota, the mode of delivery and type of food the mother consumes, antibiotic therapy, and stress [2]. Moreover, many studies have shown that an imbalance in the intestinal microbiota—dysbiosis—can lead to the development of allergic or autoimmune diseases (e.g., inflammatory bowel disease, type 1 diabetes, among others), cancer, and psychiatric disorders [3]. As such, therapeutic strategies and preparations that affect the composition of the microbiota, and thus, the patient’s well-being, have become increasingly popular.

As summarized in Figure 1, there are currently three main ways in which the microbiota can be modulated, i.e., through the use of prebiotics, probiotics, synbiotics, or postbiotics. Prebiotics are used by microorganisms as food, and, at the same time, can exert a beneficial effect on the health of the host. Currently available prebiotics include human milk oligosaccharides (HMO), lactulose, and inulin derivatives. In contrast, probiotics directly impact the gut microbiome through the selective delivery of beneficial microorganisms to the gastrointestinal tract. According to the 2002 World Health Organization (WHO) definition, probiotics are live microorganisms administered in the appropriate amounts, which have a positive effect on host health. In clinical practice, the most commonly used probiotics are bacteria of the genera *Lactobacillus*, *Bifidobacterium*, and *Streptococcus*, as well as yeast *Saccharomyces*. Despite several meta-analyses confirming the clinical effectiveness of probiotics in various diseases (including acute gastrointestinal infection and inflammatory bowel diseases [4,5]), individual reports are increasingly undermining their effectiveness and safety, especially in high-risk patients [6]. Therefore, there is increasing interest in a surrogate group of preparations: Postbiotics.

The concept of postbiotics is based on the observation that the beneficial effects of the microbiota are mediated by the secretion of various metabolites. However, its precise definition remains under discussion. According to Tsilingiri et al., postbiotics include any substance released by or produced through the metabolic activity of the microorganism, which exerts a beneficial effect on the host, directly or indirectly [7]. For the purposes of this article, we assume that postbiotics include all substances of bacterial or fungal origin that confer beneficial effect to the host and do not meet the definition of a probiotic and are not exclusively of a prebiotic nature (Figure 2).

According to the current literature, postbiotics are not considered as synbiotics. Synbiotics are a combination of prebiotics and probiotics that are claimed to have a beneficial impact on gut microbiome. However, it is believed that postbiotics may also strengthen the intestinal microbiome [8], so we believe that term “synbiotics” should be reviewed and postbiotics should be incorporated in its definition.

Although postbiotics do not contain live microorganisms, they show a beneficial health effect through similar mechanisms that are characteristic of probiotics while minimizing the risks associated with their intake. Therefore, like prebiotics, postbiotics appear to lack serious side effects while maintaining similar effectiveness to probiotics (although currently there are no studies directly comparing substances belonging to both groups).

Here, we provided a critical review of the postbiotic drugs described in the literature, including their mechanisms of action, clinical characteristics, and potential therapeutic applications.

## 2. Currently Available Classes of Postbiotic Drugs

### 2.1. Cell-Free Supernatants

Cell-free supernatants containing biologically active metabolites secreted by bacteria and yeast into the surrounding liquid can be obtained directly from cell cultures. After an incubation period, the microbes are centrifuged and then removed. Finally, the resulting mixture is filtered to ensure sterility.

Supernatants produced from cultures of different microorganisms show differing activities. *Lactobacillus acidophilus* and *Lactobacillus casei* supernatants have anti-inflammatory and antioxidant effects on intestinal epithelial cells, macrophages, and neutrophils by reducing the secretion of the pro-inflammatory tumor necrosis factor α (TNF-α) cytokine and increasing secretion of the anti-inflammatory cytokine interleukin 10 (IL-10) [9]. Meanwhile, supernatants derived from *L. casei* and *Lactobacillus rhamnosus GG* cultures can prevent the invasion of colon cancer cells [10]. As cell-free supernatants can reduce oxidative stress in vivo [11] and provide direct antitumor activity, they may be clinically useful in the prevention of cancer.

Supernatants derived from bacterial cultures of the genera *Lactobacillus* and *Bifidobacterium* were also recently shown to display antibacterial activity by preventing the invasion of enteroinvasive *E. coli* strains into enterocytes in vitro [12]. Although these antibacterial properties may result from the inhibition of adhesion of the pathogenic bacterial strains (due to competition for receptor sites), the cell supernatants could also have a local effect on the intestinal environment, cell barrier, and expression of protective genes [12]. Therefore, cell-free bacterial supernatants are promising anti-infectious agents, for example, for the treatment of diarrhea.

Meanwhile, *Lactobacillus plantarum* supernatants were found to have a positive effect on the maturation and morphological structure of the intestinal barrier [13]. Administering these supernatants to lambs early in their life was associated with an increase in the absorption surface of the intestine and a decrease in the population of intestinal pathogens [13]. The concentration of inflammatory markers (IL-1β and TNF-α) in the intestinal mucosa also decreased (*p* < 0.05) [13].

There is also evidence of the beneficial effects of supernatants derived from yeast cultures. In particular, supernatants from *Saccharomyces cerevisiae* and *Saccharomyces boulardii* reversed the state of disturbed intestinal peristalsis caused by stress stimuli [14]. *S. boulardii* supernatants also show anti-inflammatory and antioxidant activity [9], similar to bacterial cell supernatants, and can accelerate wound healing and regeneration of the intestinal barrier [15].

### 2.2. Exopolysaccharides

During their growth, microorganisms produce biopolymers with different chemical properties. These biopolymers can be released outside the bacterial cell wall, forming a heterogeneous group of substances called exopolysaccharides (EPSs). EPSs are currently used in the food industry as stabilizing, emulsifying, and water-binding agents [16], although their biological function is not entirely clear. Nonetheless, the use of EPSs in pharmaceutical products and functional foods has attracted recent interest.

EPSs may modulate the immune response by interacting with dendritic cells (DCs) and macrophages and enhancing the proliferation of T and NK lymphocytes [17]. In addition, an EPS isolated from tofu, which is a product of *L. plantarum*, induced nitric oxide (NO) secretion and enhanced the phagocytic potential of macrophages in an in vitro model [18]. This EPS also increased IgA concentrations in the intestinal mucosa (*p* < 0.05) and stimulated lymphocyte proliferation (*p* < 0.01) [18]. Meanwhile, using an EPS derived from L. casei as an adjuvant increased the effectiveness of the foot-and-mouth disease vaccine [19].

Some EPSs produced by *Lactobacillus* strains isolated from fermented Durian fruit possess antimicrobial and antioxidant properties [20]. The ability to bind iron ions was shown to account for the antioxidant potential of an EPS obtained from *Lactobacillus helveticus* called uronic acid, which, notably, is also responsible for the antioxidant properties of green tea [21].

EPSs can also have a positive effect on lipid metabolism by inhibiting cholesterol absorption [20]. Indeed, the consumption of kefiran (an EPS produced by *Lactobacillus kefiranofaciens*) delayed the development of atherosclerosis in a preclinical animal (rabbit) model (*p* < 0.05) [22]. Kefiran also prevented blood pressure increases and stabilized blood glucose levels in rats who consumed excessive cholesterol [23]. Thus, EPSs such as kefiran are potential candidates for preventing cardiovascular diseases.

β-glucans, another class of EPSs, can interact with Dectin-1 receptors on the surface of macrophages and activate them [24]. As a result, β-glucans may enhance the cellular immune response against bacteria, viruses, parasites, and cancer cells [25,26]. β-glucans may also have a positive effect on probiotics’ efficacy, for example, by facilitating the adhesion of lactobacilli to the intestinal epithelium [27]. They can also increase the bioavailability and absorption of carotenoids (compounds with antioxidant and anti-inflammatory properties) in the gastrointestinal tract [28]. Moreover, topically applied β-glucans (in the form of a cream) may help patients suffering from atopic dermatitis, by reducing the number and severity of exacerbations (*p* = 0.038) while maintaining a good safety profile [29].

### 2.3. Enzymes

Microorganisms have evolved defense mechanisms against the harmful effects of reactive oxygen species (ROS), which can damage lipids, proteins, carbohydrates, and nucleic acids. In particular, antioxidant enzymes, such as glutathione peroxidase (GPx), peroxide dismutase (SOD), catalase, and NADH-oxidase, play key roles in combating ROS. Indeed, two strains of *L. fermentum* were found to have a high content of GPx [30], and were later documented to possess potent antioxidant properties in vitro [31]. Antioxidant properties of postbiotics derived from *Lactobacillus plantarum* were demonstrated in the study conducted by Izuddin et al. [32]. This effect was observed thanks to increased GPx concentration in serum (*p* < 0.05). Moreover, genetically modified *Lactobacillus* strains that synthesize SOD or catalase showed superiority in relieving symptoms in a mouse model of Crohn’s disease relative to their unmodified counterparts [33]. Furthermore, *Lactobacillus* strains with increased catalase activity were more effective in relieving inflammation in a mouse model of inflammatory bowel disease than strains of the same bacterium producing SOD (both strains decreased the body temperature comparing to the controls with *p* < 0.05) [34]. This trial revealed that anti-inflammatory activity of *Lactobacillus* strains is dependent on the antioxidative enzyme expression profile of each strain. Besides, genetically modified *Lactobacillus lactis* expressing catalase was shown to prevent chemically induced colon cancer in mice [35]. Currently, we lack data regarding the use of sole antioxidant enzymes in vivo.

### 2.4. Cell Wall Fragments

Many components of the bacterial cell wall are immunogenic (i.e., elicit a specific immune response), including bacterial lipoteichoic acid (LTA). LTA is found in the cell walls of Gram-positive bacteria and can be spontaneously released into the environment [36]. Although LTA has been shown to exhibit immunostimulatory effects [37], data on its activity are ambiguous. Some reports indicate that LTA reduces IL-12 production and induces the production of cytokines with immunoregulatory activity (e.g., IL-10) [38]. In contrast, others have shown LTA does not alleviate inflammatory processes and actually causes damage to tissues in the intestine [39].

The use of LTA in dermatological diseases is slightly less controversial. The topical application of LTA enhances non-specific defense mechanisms, leading to the release of anti-infectious peptides, including human β-defensin and cathelicidin [40]. In fact, bacteria of the genera *Lactobacillus* and *Bifidobacteria*, which produce significant amounts of LTA, stimulate the skin mast cell response against some bacterial and viral infections [41]. This data suggests LTA may be useful for treating a wide range of skin infections. Moreover, based on its anti-inflammatory and anti-cancer activity, LTA may have broader utility [42,43]. Despite these beneficial activities, LTA may exert side effects in living organisms and cause an excessive inflammatory response. Therefore, further safety evaluation for LTA is warranted.

### 2.5. Short-Chain Fatty Acids

Short-chain fatty acids (SCFAs) are a product of fermentation of plant polysaccharides by intestinal microbiota. Well-known SCFAs include acetic, propionic, and butyric acids, which can form the corresponding fatty acid salts (i.e., acetate, propionate, and butyrate). Butyrate is one of the most important energy sources for enterocytes, as it helps to renew the intestinal epithelium and can also modulate gene expression by incompetently inhibiting histone deacetylases. Butyrate also shows immunosuppressive effects [44]. For example, butyrate has been shown to induce food tolerance by increasing the expression of immunosuppressive cytokines (e.g., type 1 interferons [IFNs], IL-10, TGF-β) and downregulating several cytokines and proinflammatory receptors (e.g., toll like receptor (TLR)2/4, Caspase-1, NLRP3, IL-1β, IL-18, IL-33, IL-25, MAPK). These immunosuppressive effects of butyrate result from the inhibition of NF-κB1 transcription factor activity and its intracellular pathways [44]. Indeed, rectal administration of butyrate caused a significant regression of inflammatory changes in the large intestine of patients with ulcerative colitis relative to patients receiving placebo [45].

Intestinal colonization with *Roseburia intestinalis*, which produces significant amounts of butyrate, was shown to inhibit atherogenesis in a mouse model of atherosclerosis [46]. The same study also noted a significant reduction in endotoxemia (likely resulting from sealing of the intestinal barrier) and inflammatory markers in the serum and aorta (including lipopolysaccharide and TNF-α) [46]. It is worth highlighting that intestinal administration of tributyrin results in analogous effects as observed in abovementioned part of the trial, implying the beneficial effect of *R. intestinalis* is mediated, at least in part, by butyrate. In terms of its potential mechanism, SCFAs can affect energy management by stimulating G-protein coupled receptors (GPCRs) and secretion of the glucagon-like peptide 1 (GLP-1). Indeed, an increase in serum and fecal acetate was associated with an increase in insulin sensitivity and a reduction in body fat in vivo, likely through increased GLP-1 levels [47]. Acetate can also directly regulate appetite in the central nervous system [48], indicating a potential application in preventing cardiovascular diseases. Furthermore, using a diet high in acetate significantly increased resistance to enterohaemorrhagic E. coli O157:H7 infection in a mouse model [49]. This phenomenon is likely a result of the sealing properties of acetate on the intestinal barrier, which prevents lethal toxins from entering the general circulation [49].

Propionate is another SCFA that is one of the main substrates of gluconeogenesis in the liver. In addition to its role in carbohydrate metabolism, propionate has a statin-like effect, inhibiting the cholesterol synthesis pathway [50]. Propionate also shows an anti-inflammatory activity effect in vivo that is comparable to that of butyrate [51]. Indeed, many studies are currently underway regarding the therapeutic use of SCFAs in medicine. For example, a recent report showed that SCFAs may provide symptom relief using an animal model of inflammation and demyelination that occurs in the brain during multiple sclerosis [52].

### 2.6. Bacterial Lysates

Bacterial lysates (BLs) are obtained by the chemical or mechanical degradation of Gram-positive and Gram-negative bacteria commonly found in the environment. Their clinical use is based on the concept of the gut-lung axis, i.e., the functional connection between the immune system of the intestine and the respiratory system [53]. In particular, studies have shown that orally administered lyophilized BLs reach the Peyer’s patches in the small intestine, where they stimulate DCs, and subsequently activate T and B lymphocytes [54]. Mature lymphocytes then migrate to the mucous membrane of the respiratory tract and initially stimulate the innate immune system and promoting IgA secretion [54]. Indeed, the safety of BLs use has been confirmed during many clinical studies on various diseases, including recurrent upper respiratory tract infections in children [55].

A causal relationship between a decrease in the incidence of infections in highly developed countries and an increase in allergic diseases has been proposed, potentially due to the so-called hygiene hypothesis. Therefore, using BLs, which mimic the presence of bacteria, to stimulate the immune system, is an attractive option in the case of insufficient exposure to microorganisms. Indeed, a 2018 meta-analysis including over 4800 children showed a significantly lower incidence of respiratory infections in those receiving a commercially available BL preparation compared to the control group [56]. Similarly, a 2020 systematic review proved the effectiveness of BL add-on therapy in reducing the frequency of wheezing episodes and asthma exacerbations in children (*p* < 0.001 for both endpoints) [57]. Infection prevention is one, though not the only, reason for the positive effects of BLs on reducing episodes of exacerbation of asthma in children [58] and chronic obstructive pulmonary disease in adults [59]. BLs can also reduce the frequency of allergic rhinitis episodes [60] and alleviate the symptoms of atopic dermatitis [61]. Finally, ingesting heat-killed *Lactobacillus paracasei* may be applicable in reducing the symptoms of dry eye syndrome, which primarily arises from the long-term, repetitive exposure to blue light emitted by LED screens [62].

### 2.7. Metabolites Produced by Gut Microbiota

The gut microbiota produces an array of molecules, including vitamins, phenolic-derived metabolites, and aromatic amino acids. Due to high bioavailability, antioxidative features, and signaling properties, these substances are considered to be important contributors in host-microbiome crosstalk.

It has been demonstrated that *in situ*-produced bacterial foliate can be absorbed in the colon and incorporated into host’s tissues. Folate plays an important role in DNA synthesis, reparation, and methylation, and is also considered as an antioxidative agent. Therefore, intestinal-produced folate may exert systemic function. Citizens of countries with mandatory folate food fortification were reported to have lower risk of stroke compared to the controls (RR, 0.85; *p* = 0.004) [63]. However, the impact of on colorectal cancer risk takes the form of a U-shaped relationship, and the optimum folate status has not been defined [64]. Nevertheless, the issue of intestinal bacteria producing folate and their potential clinical application in maintaining optimal folate status deserves further consideration. In vitro, both folate-producing *Lactobacillus helveticus* CD6 and the intracellular cell-free extract of this strain demonstrated antioxidative activity [65].

Several bacterial strains have been shown to synthesize vitamin B12 *de novo* [66]. Supplementation of Lactobacillus acidophilus in yogurt matrix was associated with elevated vitamin B12 and folate serum levels (*p* < 0.05) and reduced prevalence of anemia (*p* < 0.01) [67].

Vitamin K is a cofactor required for the synthesis of clotting factors. Whereas the contribution of microbiome-derived to vitamin K resources has been established, locally produced vitamin K appears to have underpinning mechanisms. Vitamin K concentration in the human gut has been associated with microbiome structure. However, it did not result in the alteration of inflammation biomarkers (IL-6 and TNF-α; both *p* > 0.05) [68].

Gut microbiota is postulated to be actively involved in aromatic amino acids (AAA) metabolism. AAA, as bioactive molecules, may act on distant organs, such as the kidneys, brain, and cardiovascular system [69]. For example, the genetic modification of gut microbiota metabolism enabled to control indoxyl sulfate plasma levels. Indoxyl sulfate contributes to the progression of chronic kidney disease, implying the possible role of targeting AAA metabolism in renal disorders [70].

The interplay between dietary polyphenols and gut microbiota has drawn a great deal of attention. Polyphenols modulate the structure and are concurrently metabolized by gut microbiota. Hosts’ responses to the dietary interventions may differ and this observation formed the basis for developing term called “metabotype” [71]. Metabotyping investigates the relationship between a metabolic phenotype and gut microbiome-derived metabolites that characterize the metabolism of the parent compound, thus providing a rationale for developing “personalized nutrition” [72]. Postbiotics derived from dietary polyphenols include, i.e., urolithin A (UA), equol, and 8-prenylnaringenin (8-PN). Mice treated with UA for 10 weeks weighed 23.5% less than controls. Apart from anti-obesity effects, UA improved the insulin resistance score (HOMA-IR) in a statistically significant manner (*p* < 0.001) [73]. The first human trial demonstrated the safety profile of orally administered UA, along with improvement in fatty acid oxidation rate, systemic mitochondrial health, and serum acylcarnitinen concentration (*p* < 0.05) [74]. One year of equol supplementation in middle-aged Japanese women resulted in arterial stiffness reduction (*p* < 0.001) and elevation of lipid parameters (HDL, LDL, and total cholesterol concentrations—all *p* < 0.01) [75]. Moreover, one year of equol intake caused a significant increase in whole body bone mineral density among postmenopausal women (1.040 g/cm^2^ vs. 0.994 g/cm^2^; *p* = 0.015) [76].

## 3. Potential Mechanisms of Postbiotic Action

Due to the high heterogeneity of substances classified as postbiotics and the limited framework of this study, this section summarizes only the most important mechanisms of action that are characteristic of postbiotics (Figure 3). It should be emphasized that there is currently insufficient data available to understand the complex effects of postbiotics in their entirety. However, postbiotics will likely have pleiotropic effects on the human body.

### 3.1. Immunomodulatory Effects

The immunomodulatory effects of the gut microbiome have long been suggested [77]. For example, butyrate (a SCFA) induces the differentiation of regulatory T cells (Tregs) in the intestine [78]. In addition, propionate (another SCFA) enhances the formation of peripheral Tregs [79]. Various fractions of postbiotics isolated from *Bacillus coagulans* culture (supernatant, cell wall fragments) also induce anti-inflammatory cytokine production and promote T helper (Th)2-dependent immune responses [80]. Moreover, numerous in vitro experiments have shown that the supernatant from a *Bifidobacterium breve* culture induces the maturation and survival of DCs, and consequently, increases IL-10 secretion and inhibits TNF-α secretion [81]. These properties may be responsible for limiting the Th1-mediated responses and enhancing Th2-mediated responses [82], as is often observed in those prone to atopic diseases. In a mice model, postbiotics derived from *Streptococcus thermophilus* were shown to enhance Th1 lymphocyte response in mesenteric lymph nodes compared to controls (*p* < 0.05) [82].

### 3.2. Antitumor Effects

As inflammation is inextricably linked to carcinogenesis, any substance that inhibits inflammation may also have anti-cancer potential. Indeed, the SCFA propionate (produced by *Propionibacterium freudenreichii*) was shown to selectively induce apoptosis in gastric cancer cells [83]. SCFAs also influence the regulation of oncogenes and suppressor genes through epigenetic modifications. *L. rhamnosus GG* supernatant increased ZO-1 expression (responsible for the correct structure of tight junctions between cells and cell adhesion) and decreased MMP-9 expression (which helps degrade the intercellular matrix, thus facilitating cancer cell penetration) [10]. Indeed, changes in ZO-1 and MMP-9 level caused by exposure to the *L. rhamnosus GG* supernatant helped reduce colorectal tumor cell invasion in an in vitro model (*p* = 0.01) [10].

### 3.3. Infection Prevention

Some postbiotics can have direct antimicrobial effects by sealing the intestinal barrier, competitively binding to receptors required by some pathogenic bacteria, changing the expression of host genes, or modulating the local environment [12]. Indeed, combining postbiotics and probiotics effectively prevented rotavirus-associated diarrhea in a preclinical model [84]. Furthermore, randomized clinical trials conducted in a group of children aged 12–48 months showed that daily intake of products containing *L. paracasei* postbiotic led to a reduction in the incidence of diarrhea [85,86], acute gastroenteritis, pharyngitis, laryngitis, and tracheitis [86,87]. Butyrate (a SCFA), was found to support the regeneration of the intestinal epithelium [88]. Meanwhile, supernatant obtained from one of the most common probiotic strains, *L. rhamnosus GG*, helps to protect human intestinal smooth muscle cells from damage [89]. The use of bacterial lysates in children was associated with significant reduction of respiratory tract infection compared to controls (MD = −2.33; *p* < 0.00001) [56].

### 3.4. Antiatherosclerotic Effects

Postbiotics may also play a role in lipid metabolism and could reduce the risk of cardiovascular incidents. For example, the SCFA propionate can inhibit condensation of cholesterol precursors, leading to statin-like effects [50]. Kefiran also has antiatherogenic properties, which may result from the reduction of inflammation, prevention of cholesterol accumulation in macrophages, and reduction of lipid concentration [22]. Moreover, *Lactobacillus* BLs were found to reduce the levels of triglycerides and LDL cholesterol while increasing the level of beneficial HDL cholesterol in an obese mouse model [90]. This beneficial effect of fragmented *Lactobacillus* bacteria on the lipid profile was caused by activation of the peroxisome proliferator-activated receptor (PPAR)α, which is also the therapeutic target of the fibrate class of lipid-lowering drugs [90]. Moreover, propionate stimulated the release of peptide YY and GLP-1 in human colonic cells and resulted in a significant reduction of total adipose content (*p* = 0.027) and intrahepatocellular lipid content (*p* = 0.038) in vivo [91].

### 3.5. Autophagy

Autophagy is a homeostatic mechanism through which damaged organelles and proteins are cleaned out. This self-degradative process can act as a response to various stress stimuli, including nutrient stress. Intracellular receptor NOD1 detects bacterial peptidoglycan and promotes autophagy and inflammatory signaling. Irving et al. revealed that this effect was mediated by outer membrane vesicles—spherical membrane structures naturally shed by all Gram-negative bacteria as part of their normal growth in vitro and in vivo [92]. Postbiotic-obtained *Lactobacillus fermentum* triggers autophagy in hepatic cells HepG2. Autophagy inductive potential of *L. fermentum* displayed protective effects in pharmacologically induced liver toxicity [93]. Mitophagy is a specific autophagy elimination of damaged mitochondria. In humans, urolithin A inhibits mitophagy, and may therefore prevent or delay the development of an age-related decline in muscle health [74].

### 3.6. Accelerated Wound Healing

Oxytocin is a multidirectional neuropeptide that plays a dominant role in stimulating uterine contractions during labor, modulating behavior, and creating an emotional bond. In addition, oxytocin can stimulate and accelerate wound healing. The administration of BLs obtained by sonication of *Lactobacillus reuteri* increased the number of oxytocin-producing cells in the hypothalamic periventricular nuclei, resulting in an elevated oxytocin concentration in blood serum in animal models (*p* < 0.001) [94]. Comparable results were obtained by the administration of *L. reuteri* probiotics in both animal and human models, suggesting that the use of BLs is sufficient to achieve satisfactory results, with a significantly improved safety profile [94].

## 4. Clinical Utility

### 4.1. Production Technology

Bacterial culture and the production of probiotics are somewhat unpredictable in nature. The problem of dose standardization, which is a significant issue in the production of probiotics, does not exist in the case of postbiotics. From an economic standpoint, the benefits of postbiotics include longer shelf life, easier storage, transport, and a reduced need to maintain a low temperature in comparison to probiotics. The use of a repetitive production process and the possibility of more precise quantitative control (except for BLs) are additional advantages of postbiotics compared to probiotics.

### 4.2. Safety of Use

When discussing the therapeutic benefits, attention should be paid to the superiority of postbiotics over postbiotics in the context of safety. The undoubted advantage of postbiotics is bypassing the problem of acquiring antibiotic resistance genes and virulence factors, which may occur in vivo when probiotics are used [95]. Postbiotics eliminate the need for exposure to live microorganisms, which is particularly important in children with an immature immune system and a leaky intestinal barrier.

### 4.3. Functional Food

Functional foods can be defined as dietary items with additional health benefits besides their nutritional value. Physiologic profits of functional foods are provided by adding new (e.g., probiotics or postbiotics) or already present ingredients. The favorable safety profile of postbiotics makes them rational candidates for use in functional foods. When discussing the clinical use of postbiotics, one cannot ignore galactosyllactose (3’-GL), which is formed as a result of fermentation of human milk oligosaccharides (HMOs) and can therefore be classified as postbiotic. In addition to immunomodulatory activity, 3’-GL also has natural anti-inflammatory properties and improves intestinal barrier integrity [96]. As bacteria, along with their fragments and metabolites, are passed to the baby in the mother’s milk, such a complex mixture cannot obviously be replaced by a single substance. However, combining prebiotics, postbiotics, and HMOs into one preparation is a tempting concept for mapping the composition and properties of natural human milk.

In addition, functional foods could be enriched with postbiotics to increase the host’s immune activity. For example, the cell-free fraction of fermented milk prevented *Salmonella* infection in a mouse model [97]. It is worth mentioning that the effectiveness of the postbiotics used in the abovementioned study was equivalent for preparations produced on both laboratory and industrial scales. Postbiotics from *B. breve* and *Streptococcus thermophilus* are currently used in the production of functional foods (particularly for modified milk), and their efficacy assessed in randomized clinical trials. For example, *B. breve* and *S. thermophilus* postbiotics reduced the incidence of symptoms suggestive of food or inhalation allergy in the first months of life in children with a positive history of atopy, and the effect persisted after discontinuation of the preparation [98]. The use of the above postbiotics was also associated with a milder course of acute diarrhea in infants [99]. Notably, one of the active metabolites of *S. thermophilus* is the abovementioned 3’-GL [100].

### 4.4. Use in Allergic Diseases

Postbiotics are considered may be a viable therapeutic option for allergic diseases, as they can restore the balance of Th1/Th2-mediated immune responses and support maturation of the immune system. Indeed, available data support the use of postbiotics in preventing asthma/wheezing exacerbations in children [57,101]. In addition, the severity of atopic dermatitis symptoms was inversely proportional to the number of butyrate-producing bacteria in the intestine [102], and the oral intake of BLs was associated with better results of atopic dermatitis treatment in children [61]. Finally, postbiotics may have beneficial effects in food allergies. A clinical study of over 200 children showed that the presence of a rich butyrate-producing bacterial microbiota was associated with an earlier resolution of cow’s milk allergy [103].

## 5. Future Clinical Applications

Postbiotics play a vital role in the maturation of the immune system, affect barrier tightness and the intestinal ecosystem, and indirectly shape the structure of the microbiota. As such, postbiotics may be useful in treating or preventing many disease entities, including those for which effective causal therapy has not yet been found (e.g., Alzheimer’s disease, inflammatory bowel disease, or multiple sclerosis). Indeed, clinical trials aimed at modifying the microbiota of patients suffering from the abovementioned diseases are currently underway, and the first results are promising [52,104].

Postbiotics may be particularly useful in infants, as the first months of life are critical for developing the proper structure of the microbiota. As the microbiota “matures” up to about three years of age [1], any abnormalities can be associated with short- and long-term consequences (e.g., necrotizing enterocolitis and asthma, respectively) [105]. Creating the appropriate environment for the formation of the correct microbiota appears crucial for the proper development and preservation of the child’s future well-being, and postbiotics can provide such conditions.

Postbiotics may also be useful in the prevention and treatment of SARS-CoV-2 infection, as the structure and metabolic activity of the intestinal microbiome may be related to the occurrence of biomarkers predicting the severe coronavirus disease 2019 (COVID-19) course [106].

The potential value of postbiotics is not limited to therapeutic applications. Indeed, the emergence of biological doping (and its detection) is an area of interest. A recent study in mice showed that the presence of bacteria of the genus *Veillonella* in the gut, which can metabolize lactic acid to propionate, significantly increased the animals’ physical performance [107]. A similar result was obtained by the enteral administration of propionic acid, indicating the possibility of using postbiotics to modify physical fitness and the independence of the observed effect from the presence of bacteria [107].

## 6. Summary

The use of metabolites or fragments derived from microorganisms (i.e., “postbiotics”) is an attractive therapeutic and preventive strategy in modern medicine. According to current data, such postbiotics have pleiotropic effects, including immunomodulatory, anti-inflammatory, antioxidant, and anti-cancer properties. Some of these properties are even in clinical use. The boundary between probiotics and postbiotics is blurred in some trials, as their impact on the results is often not evaluated separately. We expect further research into the biological activities of these metabolites will unveil novel uses for postbiotics in medicine and beyond.

## Figures and Tables

**Figure 1 nutrients-12-02189-f001:**
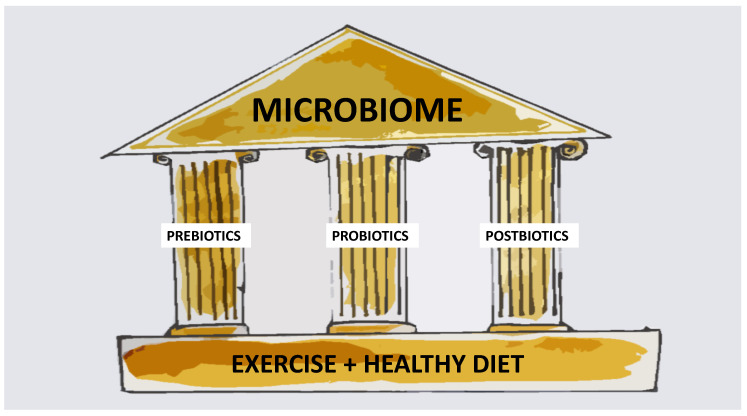
Optimal microbiome composition as a token of human wellbeing. Microbiome composition and structure is one of the factors determining proper human development and health. The roof that represents microbiome may be impermeable and reliable, only provided structures below are solid. A suitable diet and physical activity form the basis for the construction. Metaphorical foundation, which is represented by diet and exercise, underlines principal role of healthy lifestyle in sustaining human health and wellbeing. It is a lifestyle modification one should implement first when commencing the process of building human welfare. The connectors (pillars) between “roof” and “foundation” that cement the construction are pre-, pro-, and postbiotics. A lot of research is currently focused on determining the ideal proportion and shape of each pillar, so all of the construction’s elements depicted in this figure would be in the state of harmony. It is important to note that the composition of the microbiome is also affected by other factors that are not found in this figure, e.g., route of labor, use of medicines, or having siblings.

**Figure 2 nutrients-12-02189-f002:**
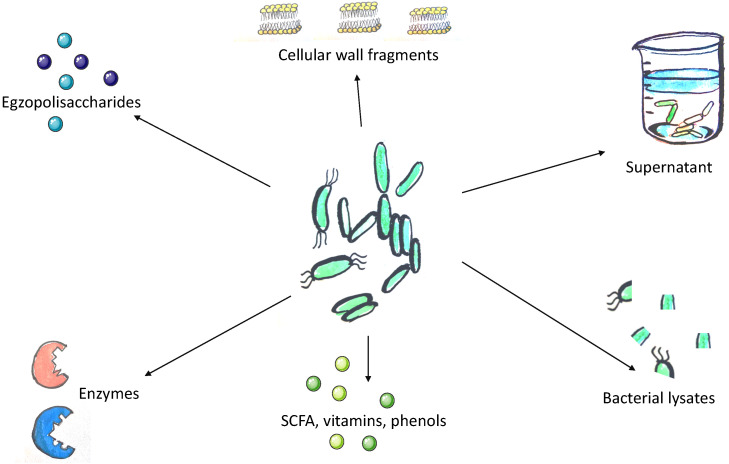
Methods of acquisition of postbiotics. The concept of postbiotics conceals either metabolites or fragments of microorganisms which confer a beneficial effect to the host. The structural heterogeneity of postbiotics implies the abundance of possible techniques used to postbiotics’ acquisition. Lysis of bacterial cells may be achieved by chemical and mechanical techniques. These methods include enzymatic extraction, solvent extraction, sonication, and heat. Extraction, dialysis, and chromatography are used to isolate and identify desired molecules. SCFA, short-chain fatty acids.

**Figure 3 nutrients-12-02189-f003:**
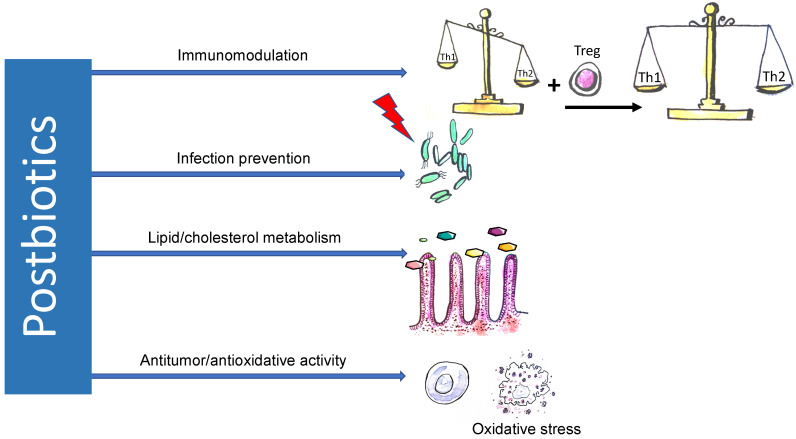
Mechanisms of action of postbiotics. Postbiotics display pleiotropic properties. Due to the induction of differentiation of T regulatory lymphocytes and synthesis of anti-inflammatory cytokines, postbiotics restore the imbalance between two major arms of immune system represented by Th1 and Th2 lymphocytes. The balance between Th1 and Th2 lymphocytes is vital for immunoregulation, and its disturbance causes various immune diseases, including atopic disorders. Antibacterial activity is probably mediated by postbiotics’ impact on the molecular structure of enterocytes, which results in sealing the intestinal barrier. “Statin-like” activity of postbiotics and its future therapeutic application in metabolic and related diseases is highly anticipated.

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
