# Peer review of "Postbiotics—A Step Beyond Pre- and Probiotics"

_nutrients, 2020, doi:10.3390/nu12082189_

Round 1

Reviewer 1 Report

This review defines Postbiotics as any substance released by or produced through the metabolic activity of the microorganism, which exerts a beneficial effect on the host, directly or indirectly and which do not live microorganisms.  They infer that these are a new class of food preparations. 

The review is well written and the text is clear.  It is also appropriate length. I have a few minor suggestions to clarify the text for the reader.

  1. I would suggest that they need to clarify the novelty of the concept as the literature has been referring to postbiotics for almost a decade. Granted, there does seem to be an increased interest in the topic so the review is timely. Additionally, they should clarify the difference between postbiotics and functional foods as well describe their relationship to synbiotics.
  1. They define Anitoxidant enymes as postbiotics. The main example cited was an In Vitro assay of cell lysates and I would not consider this as a postbiotic food.
  1. They also define SCFA as postbiotics.SCFA are natural occurring in many foods so it is hard to classify them as postbioitcs.  Additonally, the example given of the Intestinal colonization with Roseburia intestinalis is clearly a probiotic affect derived from live bacteria so the distinction between probiotic and postbiotic is somewhat confounded in this instance.
  1. The also discuss Functional foods. The difference between functional foods and postbiotics is confounded.  They give one example as Human milk oligosaccharides.  HMO are in fact actually fermented in situ by lactobacillis in the mammary gland and would be considered either probiotic by their definition.  Note, this is why we pasteurize cow’s milk to stop the fermentation process and fermented milk products have been used by mankind for millennia. 

Reviewer 2 Report

There is a growing interest in the identification and understanding of the human health benefits exerted by postbiotics as they have many attractive properties compared to pre- and pro-biotics. This review article by Żółkiewicz et al detailed the pleiotropic effects of postbiotics associated with immunomodulatory, anti-inflammatory, antioxidant and anti-cancer activities. The review may contribute to the field but needs to be written more comprehensively. 

Specific comments are:

  1. Section 2. Authors reviewed a number of postbiotics including cell-free supernatants, SCFA, EPS, enzymes, cell wall fragments and bacterial lysates but missed vitamins/co-factors, organic acids, as well as dietary phenolic-derived molecules or metabolites produced by the gut microbiota.
  2. Are the postbiotics induced changes in bioactivity, biomarker measurements, etc. statistically significant? Only a few studies reviewed included p values. More details in study design may be included.
  3. Section 3.2. In addition to the inhibition of carcinogenesis, inflammation and protein expressions as reviewed, reducing cell viability, activating pro-apoptotic cell death pathways, increasing apoptosis and necrosis, increasing tumor cell death via autophagy and anti-proliferative activity by postbiotics have been reported that involve inactivated cells, cell-wall constituents, peptidoglycan and cytoplasm extracts.  
  4. Section 3, while most of the studies were reported based on the results from in vitro models, preclinical studies on animals and human interventions would be more relevant.
